# Dual loss of human *POLQ* and *LIG4* abolishes random integration

Shinta Saito[1], Ryo Maeda[2] & Noritaka Adachi[1,3]

Homologous recombination-mediated gene targeting has greatly contributed to genetic analysis in a wide range of species, but is highly inefficient in human cells because of overwhelmingly frequent random integration events, whose molecular mechanism remains elusive. Here we show that DNA polymerase θ, despite its minor role in chromosomal DNA repair, substantially contributes to random integration, and that cells lacking both DNA polymerase θ and DNA ligase IV, which is essential for non-homologous end joining (NHEJ), exhibit 100% efficiency of spontaneous gene targeting by virtue of undetectable levels of random integration. Thus, DNA polymerase θ-mediated end joining is the sole homology-independent repair route in the absence of NHEJ and, intriguingly, their combined absence reveals rare *Alu-Alu* recombination events utilizing a stretch of homology. Our findings provide new insights into the mechanics of foreign DNA integration and the role of DNA polymerase θ in human genome maintenance.

[1] Department of Life and Environmental System Science, Graduate School of Nanobioscience, Yokohama City University, Yokohama 236-0027, Japan. [2] Department of Biology, Graduate School of Science, Chiba University, Chiba 263-8522, Japan. [3] Advanced Medical Research Center, Yokohama City University, Yokohama 236-0004, Japan. Correspondence and requests for materials should be addressed to N.A. (email: nadachi@yokohama-cu.ac.jp).

Since the first reports in the mid-1980s (refs 1,2), efficient gene targeting has been hampered by random integration (RI), which typically occurs several orders of magnitude more frequently than homologous recombination (HR)-mediated targeted integration[3]. In several fungi, inactivation of the non-homologous end joining (NHEJ) pathway results in ~100% gene targeting by virtue of undetectable levels of RI[4], but unfortunately this is not the case in mammalian cells; for example, loss of DNA ligase IV (Lig4), which is indispensable for NHEJ[5], can reduce RI of foreign DNA, but does not reduce RI of targeting vectors harbouring long homology arms[6–8]. Thus, the exact contribution of NHEJ to RI is still uncertain and undoubtedly an NHEJ-independent route for RI does exist.

NHEJ and HR are the two major pathways for repairing DNA double-strand breaks (DSBs)[9,10]. NHEJ absolutely requires the Ku70–Ku80 complex and the Xrcc4 (X-ray repair cross-complementing 4)–Lig4 complex to fix DSBs, with the aid of additional factors such as DNA-dependent protein kinase catalytic subunit (DNA-PKcs), Artemis, DNA polymerases μ and λ, XLF (Xrcc4-like factor)/Cernunnos and PAXX (paralogue of Xrcc4 and XLF) when the DSB is not readily ligatable[11–13]. NHEJ-mediated repair is more error-prone than HR, because it accompanies frequent nucleotide loss and addition[5]. In addition to NHEJ, mammalian cells possess at least one genetically distinct pathway for homology-independent repair termed alternative end-joining (A-EJ)[14,15], which is considered suppressed by NHEJ and HR[16–18]. DSB repair by A-EJ is even more inaccurate than NHEJ, as it typically involves end resection of the DNA ends that is reliant on carboxy-terminal binding protein interacting protein (CtIP) and the Mre11 (Meiotic recombination 11)-Rad50-Nbs1 (Nijmegen breakage syndrome protein 1) complex[19,20], although this end resection is not as extensive as that required for initiating HR or single-strand annealing (SSA)[13,21]. SSA represents another homology-dependent repair[22], which is potentially deleterious to genome integrity, particularly when the genome contains many repetitive DNA sequences, as exemplified by Alu elements that are present at >1 million copies in the human genome and comprise ~11% of human DNA[23,24].

Although the exact molecular mechanism of A-EJ is still less characterized, accumulating evidence implicates DNA polymerase θ (Pol θ) in A-EJ[25–27]. Pol θ is a proofreading-deficient A-family DNA polymerase and is encoded by the POLQ gene[28–30]. Pol θ is unique among DNA polymerases as it possesses a helicase-like domain at its amino terminus[31]. Several recent studies have shown that loss of Pol θ results in decreased DSB repair activity[32–34]; however, it remains an open question whether Pol θ explains all non-homologous DSB repair other than NHEJ. Also unexplored is whether Pol θ contributes to RI of transfected DNA.

In this study, we show that human Pol θ is indispensable for A-EJ, which is the sole homology-independent DSB repair route in the absence of NHEJ and is typically characterized by either 2–6 bp microhomology or templated insertions at the junctions. Remarkably, despite its minor role in chromosomal DSB repair, Pol θ substantially contributes to RI and cells doubly deficient in Pol θ and Lig4 exhibit 100% gene-targeting efficiency because of virtually no RI events. Further, we provide the first direct evidence that the dual loss of Pol θ and Lig4 permits, albeit infrequent, SSA-type recombination between two separate Alu elements. Our findings provide new insights into the mechanics of foreign DNA integration and the role of Pol θ in preserving genome integrity.

## Results

**Characterization of Lig4-independent RI events.** To investigate the mechanism of Lig4-independent RI, we used a promoterless targeting vector, p4.5 HPRT-2A-EGFP-2A-Puro[35], to perform gene targeting in LIG4-knockout cells derived from a human diploid cell line, Nalm-6 (refs 36,37), and sought to characterize the nature of random integrants ('RI clones') arising in an NHEJ-independent manner (Fig. 1a and Supplementary Fig. 1). Sequence analysis of vector-genome junctions of those RI events revealed that 15 out of 26 (58%) utilized 2–6 bp microhomology for joining with no insertions (Fig. 1b–d and Supplementary Data 1). The remaining 11 junctions were associated with insertions,

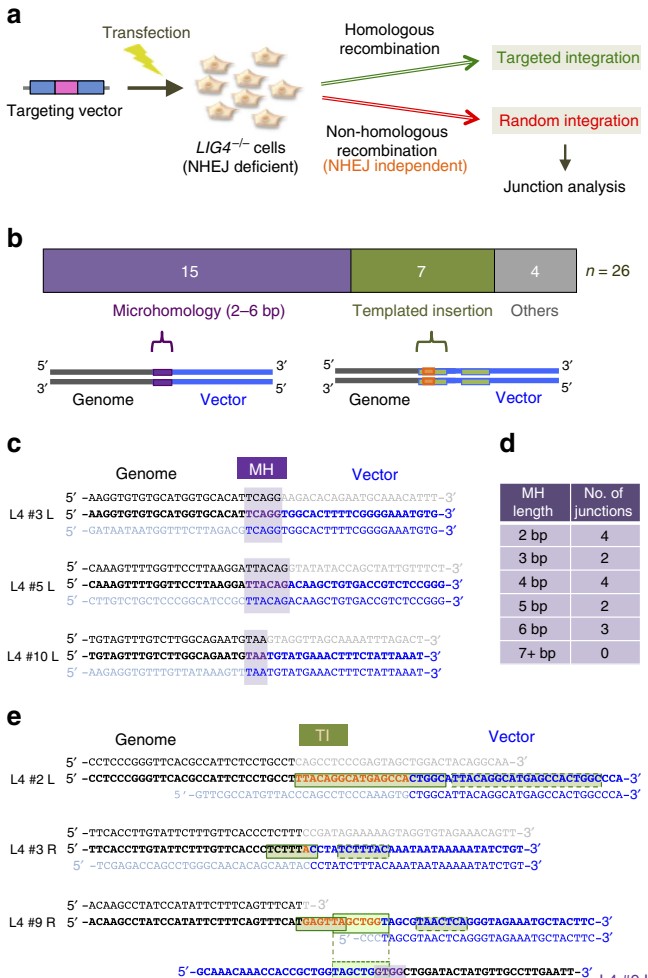

**Figure 1 | Characterization of Lig4-independent RI events.** (a) Scheme for analysis of Lig4-independent RI. A promoterless targeting vector, p4.5 HPRT-2A-EGFP-2A-Puro, was transfected into human LIG4-knockout cells and the junctions of RI clones were analysed (see Supplementary Fig. 1). (b) Features and distribution of Lig4-independent junctions. The features were classified into three categories; microhomology (2–6 bp homology), templated insertion (≥6 bp direct or inverted repeats) and others showing 1–10 bp of undefined insertions. (c) Representative junction sequences relying on microhomology. Shown are the 5′ (left) junctions of clones #3, #5 and #10 (see Supplementary Data 1). Genome and vector sequences deleted after the RI event are shown in grey and faint blue, respectively. (d) Distribution of lengths of microhomology observed in the 15 junctions. (e) Representative junction sequences accompanying templated insertions. Shown are the 5′-junction of clone #2 and the 3′-junctions of clones #3 and #9 (see Supplementary Data 1). Inserted nucleotides are indicated in orange. It is noteworthy that the sequence surrounded by a square is identical to a nearby sequence surrounded by a dotted-line square or, as can been seen at the 3′- (right) junction of clone #9 (bottom), to a sequence present in the other (left) side of the junctions (see Supplementary Data 1).

7 of which were characterized by inclusion of a unique feature called 'templated insertions'[25,26], whereas 4 junctions contained 1–10 bp undefined insertions (Fig. 1b). Consistent with the intrinsic complexity of RI events[38], the precise mechanisms for several junctions associated with templated insertions were difficult to interpret (Supplementary Data 1), but some junctions showed a simple pattern of templated insertions (that is, ≥6-bp direct or inverted repeats), including those copied sequences from the other side of the vector-genome junctions (Fig. 1e). As the traits of the templated insertions resemble those of Pol θ-mediated end joining recently reported in non-mammalian species[39,40], we reasoned that Pol θ should be involved in at least some, and perhaps most, of RI events occurring in *LIG4*-knockout cells.

**Loss of Pol θ and Lig4 eliminates RI.** To investigate the involvement of Pol θ in RI, we knocked out the *POLQ* gene (which encodes Pol θ) by gene targeting in *LIG4*-knockout cells (Fig. 2a and Supplementary Fig. 2a–e). The double-knockout cells grew slower, but were viable (Supplementary Fig. 2k) and showed wild-type levels of transfection efficiency (Supplementary Fig. 2l,m), thus allowing us to perform RI assay (Supplementary Fig. 3a). Astonishingly, pPGKneo-electroporated double-knockout cells gave rise to no G418-resistant colonies (Fig. 2b,c). We consistently obtained the same results when we repeated RI assay more than ten times by treating a larger number of cells ($0.8 \times 10^8$ in total) (Fig. 2b and Supplementary Fig. 3b) or by employing a Nucleofector technology for transfection (Supplementary Fig. 3c). By contrast, double-knockout cells complemented with *LIG4* cDNA were proficient in producing RI clones (Fig. 2b,c and Supplementary Figs 2f and 3c). In addition, very similar results were obtained using independently constructed *POLQ*-knockout Nalm-6 cells (Supplementary Fig. 3d).

We next performed gene-targeting assays at the *HPRT* locus using a promoter-containing targeting vector, pHPRT8.9-Neo2(−) (Supplementary Fig. 3e), which normally gives rise to a considerable amount of RI clones. Given the absence of RI events, we anticipated that all the drug-resistant colonies derived from double-knockout cells would be targeted clones and this was indeed the case (Fig. 2d). The 100% efficiency of gene targeting was consistently observed at several other loci such as the *AAVS1* safe-harbour locus, irrespective of homology-arm length or marker-gene cassette of the targeting vector (Table 1). The *LIG4*-complemented double-knockout cells as well as *POLQ*-knockout cells did not exhibit ultrahigh targeting efficiency (Fig. 2d and Supplementary Fig. 3f–h), demonstrating that the presence of either Pol θ or Lig4 is sufficient to bring about RI.

**Evidence for Pol θ-mediated RI in NHEJ-proficient cells.** We then analysed RI clones derived from p4.5 HPRT-2A-EGFP-2A-Puro-transfected *POLQ*-knockout cells. These Pol θ-independent junctions showed a totally different pattern from Lig4-independent junctions, with no apparent templated insertions and 42% of junctions (10 out of 24) showing 0–1 bp homology (Fig. 3a,b and Supplementary Data 2). Our data clearly suggest that 0–1 bp (or perhaps 0–2 bp) homology is characteristic of NHEJ-mediated RI, whereas templated insertion and 2–6 bp microhomology are Pol θ-dependent RI. These observations led us to revisit the junctions of RI events in wild-type cells. Notably, out of 60 junctions, 22 (37%) displayed templated insertions (≥6 bp repeats) and 6 (10%) relied on 2–6 bp microhomology (Fig. 3a,b and Supplementary Data 3), revealing that Pol θ-mediated end-joining does operate to cause RI even when NHEJ is functionally normal. It is interesting that whereas templated insertions appear favoured in wild-type cells, utilization of microhomology is

prominent when Lig4 is absent (Fig. 3a). We also note that in RI events that present templated insertions, the location of template sequence is not just on a genome or vector sequence present nearby but also on a sequence around the opposite junction of the integration site in both wild-type and *LIG4*-knockout cells (Fig. 3c). These findings again highlight the intrinsic complexity of RI reactions, yet it is now proven that this complexity is totally attributed to Pol θ-mediated, but not NHEJ-mediated RI.

**Loss of Pol θ and Lig4 reveals homology-mediated RI.** As we wished to further examine whether the dual loss of Pol θ and Lig4 completely abolish RI, we then employed a MaxCyte STX electroporation instrument for transfection. Transient expression assays showed that the MaxCyte system allows higher transfection efficiencies than other methods (Fig. 4a). Consistent with this expectation, RI frequencies in wild-type and *LIG4*-knockout cells were >10 times higher than those values shown above (Fig. 4b). When pPGKneo was MaxCyte transfected into double-knockout cells ($10^8$ cells), five drug (G418)-resistant colonies appeared, meaning that the RI frequency was $4.5 \times 10^{-7}$, not zero (Fig. 4b). These clones showed severe growth retardation, perhaps reflecting the occurrence of some sort of chromosomal rearrangements associated with vector integration; nevertheless, we were able to determine one junction, which intriguingly utilized a 16 bp region of homology between the vector (*Amp*[r] in the vector backbone) and the genome (chromosome 12) (Fig. 4c), suggesting an SSA-type recombination. We next performed a gene-targeting assay using the promoter-containing *HPRT* targeting vector and found that the targeting efficiency in double-knockout cells was 99.4% (1,082/1,088), not 100% (Fig. 4d). Among the six RI clones, seven junctions analysed were all found to utilize homology-directed SSA-type recombination between two *Alu* elements, one of which resides in the homology arm of the vector and the other in the genome (chromosome 2 or 12) (Fig. 4e). Remarkably, these SSA-mediated RI events rely on ≥16 bp homology, a tendency clearly distinguished from Pol θ-mediated A-EJ that favours ≤6 bp microhomology. From these results, we conclude that even though non-homologous DNA integration is completely suppressed when Pol θ and Lig4 are both missing, SSA (or some other homology-mediated recombination) is capable of bringing about a very low level of RI under certain conditions.

**Role of Pol θ in chromosomal DSB repair.** Finally, we examined the importance of human Pol θ in chromosomal DSB repair. As shown in Fig. 5a, loss of Pol θ caused only a small or no increase in cellular sensitivity to etoposide (a topoisomerase II inhibitor that induces DSBs[41]) in wild-type cells, but significantly augmented etoposide hypersensitivity of *LIG4*-knockout cells. These results indicate that Pol θ only plays a minor role in repair of DSBs and suggest that the absence of both A-EJ and NHEJ leads to severe compromise in non-homologous DSB repair. To further confirm this, we employed CRISPR-Cas9 technology[42] to induce a DSB at the *HPRT* locus and look at rejoining of the chromosomal DSB (Fig. 5b). As summarized in Fig. 5c and Supplementary Data 4, neither microhomology nor templated insertion was observed in cells lacking Pol θ ($n = 49$; with only one exception having 2 bp microhomology, which is reasonably attributable to NHEJ repair[11]), clearly showing that Pol θ is indeed indispensable for A-EJ. In sharp contrast, *LIG4*-independent junctions were associated with larger terminal deletions (Fig. 5d) and 24 out of 28 (86%) relied on microhomology (2–6 bp homology) or templated insertion (≥6 bp repeats) (Fig. 5c and Supplementary Data 4). These results show that Pol θ (and hence A-EJ) efficiently comes to the

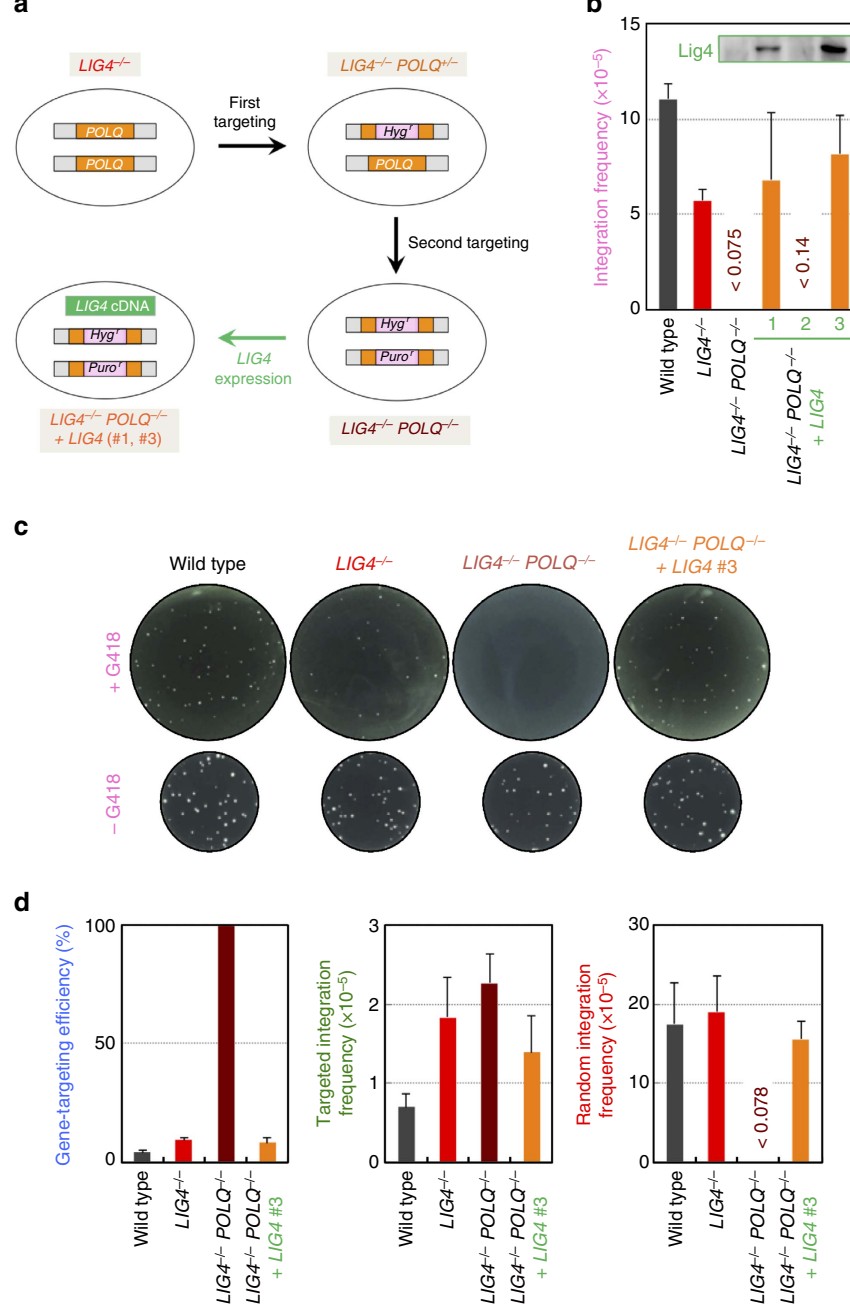

**Figure 2 | Dual loss of Pol θ and Lig4 suppresses RI.** (**a**) Scheme for construction of mutant cell lines by gene targeting. The two *POLQ* alleles of *LIG4*-knockout cells were sequentially disrupted by HR-mediated gene targeting to yield heterozygous (*LIG4$^{-/-}$ POLQ$^{+/-}$*) and homozygous (*LIG4$^{-/-}$ POLQ$^{-/-}$*) knockout cells. Ectopic expression of *LIG4* cDNA in *LIG4$^{-/-}$ POLQ$^{-/-}$* cells yielded two *LIG4*-complemented cell lines (#1, #3). It is noteworthy that clone #2 failed to express Lig4 (see the inset in **b**) and thus served as a negative control for further analysis. (**b**) Integration frequency of pPGKneo in wild-type, *LIG4$^{-/-}$*, *LIG4$^{-/-}$ POLQ$^{-/-}$* and *LIG4*-complemented *LIG4$^{-/-}$ POLQ$^{-/-}$* cells (mean ± s.d.; n = 3). Inset: western blot analysis for Lig4 (extracted from Supplementary Fig. 2f; shown for reference only). (**c**) Representative colonies arose in an experiment of **b**. Digital photographs were taken after three weeks of colony formation. Upper: G418-containing plates. Lower: G418-free plates. (**d**) Gene-targeting efficiency, targeted integration frequency and RI frequency of pHPRT8.9-Neo2(−) in wild-type, *LIG4$^{-/-}$*, *LIG4$^{-/-}$ POLQ$^{-/-}$* and *LIG4*-complemented *LIG4$^{-/-}$ POLQ$^{-/-}$* cells (mean ± s.d.; n = 3).

fore when NHEJ cannot deal with the DSB. In double-knockout cells, the efficiency of DSB joining was significantly reduced (Supplementary Data 4) and all junctions analysed accompanied large (>2 kb) terminal deletions (Fig. 5d,e), a finding consistent with earlier work in *Drosophila*[43]. Notably, these junctions all displayed clear traits of SSA-type recombination between two separate *Alu* elements (Fig. 5f), similar to that seen in rare RI clones described above.

## Discussion

Pol θ is a unique DNA polymerase in that it is highly expressed in various types of cancer[44–46]. Intriguingly, loss of Pol θ is synthetically lethal with HR defects[33,47] and thus developing Pol θ inhibitors will lead to a new therapeutic agent for various cancers with compromised HR[40,48]. Most remarkably, in this study, we have demonstrated 'synthetic abolishment of RI' by the dual loss of Pol θ and Lig4 (Supplementary Fig. 4a).

**Table 1 | Summary of gene-targeting assays in *LIG4*$^{-/-}$ *POLQ*$^{-/-}$ cells.**

| Target locus | Targeting vector | | | No. of cells treated ($\times 10^6$) | Gene-targeting efficiency |
|---|---|---|---|---|---|
| | 5′-arm | 3′-arm | Selection marker* | | |
| HPRT | 3.8 kb | 5.1 kb | P$_{PGK}$-Neo | 40.0 | 100% (138/138) |
| HPRT | 1.7 kb | 2.8 kb | 2A-Puro | 24.0 | 100% (59/59) |
| AAVS1 | 4.4 kb | 3.1 kb | P$_{PGK}$-Neo | 4.0 | 100% (8/8) |
| PRKDC | 2.1 kb | 5.1 kb | P$_{PGK}$-Hyg | 8.0 | 100% (3/3) |
| RAD54B | 3.3 kb | 4.5 kb | P$_{PGK}$-Hyg | 8.0 | 100% (5/5) |
| RAD54L | 2.9 kb | 3.6 kb | 2A-Puro | 8.0 | 100% (1/1) |
| TP53BP1 | 0.9 kb | 1.5 kb | IRES-Hyg | 8.0 | 100% (2/2) |

P$_{PGK}$, PGK promoter; 2A, 2A peptide sequence.
*It is noteworthy that although Hyg$^r$ and Puro$^r$ (hygromycin- and puromycin-resistance gene, respectively) have already been used to disrupt *POLQ*, increasing drug concentrations allowed us to reuse these genes as gene-targeting selection markers.

Tijsterman and coworkers[49] have very recently reached the same conclusion using mouse embryonic stem cells deficient in Pol θ and NHEJ. We emphasize that complete or even near-complete suppression of RI in mammalian cells is highly intriguing in light of the intrinsic complexity of RI as well as the long history of gene targeting struggling with unwanted RI clones, even despite recent advances in artificial nucleases, which stimulate RI as well as targeted integration of a donor DNA[50]. We propose that simultaneous inhibition of Pol θ and NHEJ will provide a novel strategy for ultra-efficient gene targeting even without using artificial nucleases.

Mechanistically, we have presented evidence that A-EJ depends crucially on Pol θ and this form of DSB repair is typically characterized by 2–6 bp microhomology or templated insertions at the junctions. A similar preference was recently observed in Pol θ-mediated end-joining in several mammalian cell types[51,52], consistent with a unique property of Pol θ to stabilize the annealing of two single-stranded DNA molecules with as little as 2 bp of homology[31,53]. Intriguingly, the relative usage of microhomology versus templated insertion was similar between RI and chromosomal DSB repair when Lig4 is absent (Figs 1b and 5c). Clearly, however, the importance of Pol θ in these processes appears to be distinct: a minor role in DSB repair and a substantial contribution to RI even in the presence of NHEJ (Supplementary Fig. 4b). This implies that there exists an unknown mechanism that prevents NHEJ from causing RI, allowing a larger contribution of Pol θ. This notion is well supported by our observations that A-EJ usage was marginal in chromosomal DSB repair, whereas templated insertions were frequently observed in RI clones from wild-type cells. Finally, it should also be emphasized that we have shown for the first time that the dual loss of Lig4 and Pol θ reveals SSA-type recombination that was not seen when either Lig4 or Pol θ was present. As Pol θ facilitates the formation of an annealed intermediate using as little as 2 bp of homology[31,53], more extensive resection of DNA ends would be prevented, thereby minimizing the occurrence of deleterious DSB repair by SSA. Mechanistic differences of A-EJ and SSA have long been elusive, particularly in terms of homology requirements[22]. In this study, we have shown in both RI assay and chromosomal DSB joining assay that SSA-type recombination relies on ≥13 bp homology, which contrasts with the 2–6 bp preference seen in A-EJ. Our findings described here shed light on the mechanism of SSA (between *Alu* elements), which, albeit normally suppressed, has been reported to cause deleterious outcomes involving deletions and translocations[54–56].

## Methods

**Plasmids.** Targeting vectors were constructed using the MultiSite Gateway system (Life Technologies) to assemble two homology arms and a drug-resistance gene cassette[35,37,57]. To generate targeting vectors for the human *POLQ* gene, 3.0 and 5.3 kb *POLQ* genomic fragments were PCR amplified with Tks Gflex DNA Polymerase (Takara Bio) using Nalm-6 genomic DNA as a template and were used as 5′- and 3′-arms, respectively. The primers used were: POLQ 5′ Fw (5′-GGGG ACAACTTTGTATAGAAAAGTTGGGCTCAGTACCATGTTACAGAGG-3′) and POLQ 5′ Rv (5′-GGGGACTGCTTTTTTGTACAAACTTGTAGTAGTTCC ATGTTGTGCCAGC-3′) for the 5′-arm and POLQ 3′ Fw (5′-GGGGACAGCT TTCTTGTACAAAGTGGGTGCACTACTGTGGCAGACCTTGCTAGAGC-3′) and POLQ 3′ Rv (5′-GGGGACAACTTTGTATAATAAAGTTGCTATATTACC CTGTTATCCCTAGCGTAACTCATTGGAACAAGTCAACTGCTGG-3′) for the 3′-arm. A floxed hygromycin- or puromycin-resistance gene was placed between 5′- and 3′-arms, thus yielding targeting vectors pPOLQ 2A-Hyg and pPOLQ 2A-Puro. An *AAVS1* targeting vector, pAAVS1 Neo, was constructed using the MultiSite Gateway system to assemble two homology arms (4.4 and 3.1 kb) and a neomycin-resistance gene cassette. Genomic fragments for homology arms were PCR amplified with *att*B-containing primers (AAVS1 5′ Fw (5′-GGGGACAACT TTGTATAGAAAAGTTGGCTTTGCCACCCTATGCTGACACC-3′) and AAVS1 5′ Rv (5′-GGGGACTGCTTTTTTGTACAAACTTGCAAGAAGCGCACCACCTC CAGGTTC-3′) for the 5′-arm and HPRT 3′ Fw (5′-GGGGACAGCTTTCTTGTA CAAAGTGGAACATCGCCGCCGTCAACAGTGAC-3′) and HPRT 3′ Rv (5′-GGGGACAACTTTGTATAATAAAGTTGCTATATTACCCTGTTATCCC TAGCGTAACTCAGAGGACTTTGCGGTGTTGGAGG-3′) for the 3′-arm. A *PRKDC* targeting vector, pPK-Hyg, was designed to replace exons 8–10 with a floxed hygromycin-resistance gene. Briefly, 2.1 and 5.1 kb *PRKDC* genomic fragments were PCR amplified with *att*B-containing primers (PK 5′ Fw (5′-GGG GACAACTTTGTATAGAAAAGTTGAGAGGTGGAGCACACGACCCACGGG-3′) and PK 5′ Rv (5′-GGGGACTGCTTTTTTGTACAAACTTGCAGTCCACTCTCC TGCAGCCCCTCC -3′) for the 5′-arm and PK 3′ Fw (5′-GGGGACAGCTTTCTT GTACAAAGTGGTGGTCGGCTGAGGAGACCGTGCTTG -3′) and PK 3′ Rv (5′-GGGGACAACTTTGTATAATAAAGTTGCGAGGATCTGATGGTG CACCTCTTC -3′) for the 3′-arm. A *RAD54B* targeting vector, p7.8 RAD54B Hyg, was designed to replace exons 5 and 6 with a floxed hygromycin-resistance gene. Briefly, 3.3 and 4.5 kb *RAD54B* genomic fragments were PCR amplified with *att*B-containing primers (R54B 5′ Fw (5′-GGGGACAACTTTGTATAGAAAAGTTGTT AATTAATGGTTTTATGCTAGCTCAGTGAAGC-3′) and R54B 5′ Rv (5′-GGGGA CAGCTTTCTTGTACAAAGTGGGTGGTTCTTGATAGATGCCACGTG-3′) for the 5′-arm and R54B 3′ Fw (5′-GGGGACAGCTTTCTTGTACAAAGTGGATG GACCCTACTGGAAGTCAGTTGG-3′) and R54B 3′ Rv (5′-GGGGACAACTTT GTATAATAAAGTTGCTATATTACCCTGTTATCCCTAGCGTAACTAGGCT AATGAGGGGCTGTAGTTGTC-3′) for the 3′-arm. A *RAD54* targeting vector, p6.5 RAD54 2A-Puro, was designed to replace exons 4–7 with a floxed puromycin-resistance gene. Briefly, 2.9 and 3.6 kb *RAD54* genomic fragments were PCR amplified with *att*B-containing primers (R54 5′ Fw (5′-GGGGACAACTTTGTAT AGAAAAGTTGGAATTCGGGGACTTTGAAAGGCTTTGAC-3′) and R54 5′ Rv (5′-GGGGACAGCTTTCTTGTACAAAGTGGGTGGTTCTTGATAGATG CCACGTG-3′) for the 5′-arm and R54 3′ Fw (5′-GGGGACAGCTTTCTTGTA CAAAGTGGGCCAGAGTCCAGAGTGCAAGCCAG-3′) and R54 3′ Rv (5′-GGG GACAACTTTGTATAATAAAGTTGCTATATTACCCTGTTATCCCTAGCGTA ACTAAAAGCGTTACTGGGAGGAAGATG-3′) for the 3′-arm. A *TP53BP1* targeting vector, p2.4 53BP1 IRES-Hyg, was designed to replace exon 2 with a floxed hygromycin-resistance gene. Briefly, 0.9 and 1.5 kb *TP53BP1* genomic fragments were PCR amplified with *att*B-containing primers (53BP1 5′ Fw (5′-GGGGACAACTTTGTATAGAAAAGTTGGCAGCTGTTCTCTTTACC ATGGAG-3′) and 53BP1 5′ Rv (5′-GGGGACTGCTTTTTTGTACAAACTTGT TCCAGTAGGGTCCATCTGCTCC-3′) for the 5′-arm and 53BP1 3′ Fw (5′-GG GGACAGCTTTCTTGTACAAAGTGGATGGACCCTACTGGAAGTCAGTT GG-3′) and 53BP1 3′ Rv (5′-GGGGACAACTTTGTATAATAAAGTTGCTATA TTACCCTGTTATCCCTAGCGTAACTCAAGTCCCATTTCCTAATCCAC-3′) for the 3′-arm. An *HPRT* targeting vector, pHPRT8.9-Neo2( − ), was constructed by inserting a neomycin-resistance gene cassette into the XhoI site of an 8.9 kb *HPRT* fragment[58]. The entire DNA sequence of pHPRT8.9-Neo2( − ) has been deposited in the EMBL/DDBJ/GenBank database under accession number

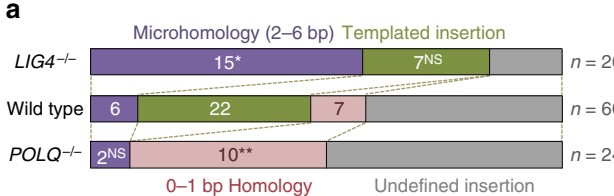

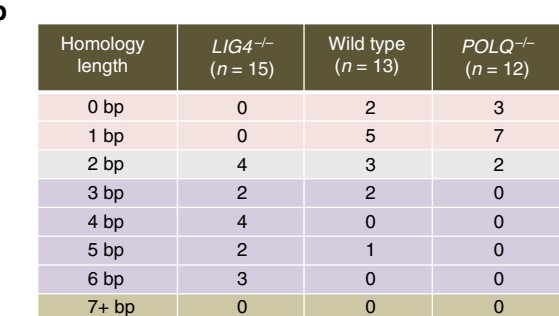

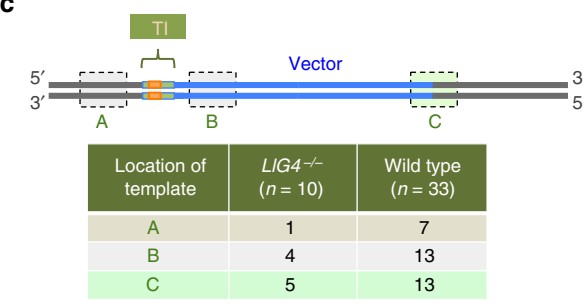

**Figure 3 | Evidence for Pol θ-mediated RI in NHEJ-proficient cells.**
(**a**) Features and distribution of junctions in RI clones from wild-type and
$POLQ^{-/-}$ cells. The junction sequences are shown in Supplementary Data
2 and 3. The features were classified into four categories; microhomology
(2–6 bp homology), templated insertion ($\geq 6$ bp direct or inverted repeats),
undefined insertion (1–11 bp) and 0–1 bp homology. The $LIG4^{-/-}$ data
(same as Fig. 1b) is shown for reference only. A Fisher's exact test was used
to determine statistical significance. NS, not significant; *$P < 0.00001$ and
**$P < 0.01$. Although not indicated, the absence of templated insertion in
$POLQ^{-/-}$ cells is statistically significant ($P < 0.0003$). (**b**) Distribution of
junctions that arose from direct (insertion-free) end joining in terms of
terminal homology length. The $LIG4^{-/-}$ data is the same as Fig. 1d.
(**c**) Distribution of templated insertion events in terms of location of the
original template sequence. The locations were categorized into three
groups: a genome sequence present near the junction (A), a vector
sequence present near the junction (B) and a sequence present at or near
the other (opposite) junction of the integration site (C); although not
shown in the figure, a vector sequence deleted after RI can also be used as a
template. It is noteworthy that $POLQ^{-/-}$ data are not presented owing to
the complete absence of templated insertion events in these cells.

LC269955. The sequence of the p4.5 HPRT-2A-EGFP-2A-Puro vector[35] is also
deposited in the database under accession number LC270125. All the targeting
vectors were purified with QIAGEN Plasmid *Plus* Midi Kits (QIAGEN K.K.) and
linearized with appropriate restriction enzyme before transfection[57].

**Cell culture.** Nalm-6 cells were cultured in a 5% $CO_2$ incubator at 37 °C in MEM
medium (Nissui Seiyaku) supplemented with 10% calf serum (Hyclone), 2 mM
MEM non-essential amino acids (Wako Pure Chemical), 1 mM sodium pyruvate
(Wako Pure Chemical), 50 μM 2-mercaptoethanol (Wako Pure Chemical) and
0.15 μM Vitamin B12 (Sigma-Aldrich). Nalm-6 is a human pre-B cell line available
from ATCC and was kindly provided by Dr Michael R. Lieber. The Nalm-6 cell line
was originally established from the peripheral blood of a 19-year-old man with
acute lymphoblastic leukaemia[59] and has a stable near-diploid karyotype[60]. All cell
lines used in this study are not listed in the database of commonly misidentified cell

lines maintained by ICLAC and were tested for mycoplasma contamination using
an e-Myco Mycoplasma PCR Detection Kit (iNtRON Biotechnology, Inc.).

**Transfection.** DNA transfection using the GTE-1 electroporation apparatus
(Shimadzu) was performed according to the manufacturer's instructions. Briefly,
cells were washed twice with Saline G (130 mM NaCl, 5.3 mM KCl, 1.1 mM
$Na_2HPO_4$, 1.1 mM $KH_2PO_4$, 6.1 mM glucose, 0.49 mM $MgCl_2$ and 0.9 mM $CaCl_2$)
and an aliquot of the cell suspension ($4 \times 10^6$ cells in 40 μl of Saline G) was
electroporated with 4 μg of plasmid DNA. The cells were then cultured for 24 h and
replated at a density of $0.5$–$1 \times 10^6$ cells per 90 mm dish into agarose medium
containing hygromycin B (0.3–0.5 mg ml$^{-1}$, Wako Pure Chemical), puromycin
(0.15–0.5 μg ml$^{-1}$, Wako Pure Chemical) or G418 sulfate (1.0 mg ml$^{-1}$, Wako
Pure Chemical). Meanwhile, small aliquots of the transfected cells were replated
into drug-free agarose medium to determine the plating efficiency. DNA trans-
fection (Nucleofection) using the Nucleofector II system (Lonza) was performed
according to the manufacturer's instructions. Briefly, $2 \times 10^6$ cells were suspended
with 100 μl of the supplied solutions (Solution T) and transfected with 2 μg of
linearized vector. DNA transfection using the MaxCyte STX device (MaxCyte) was
performed according to the manufacturer's instructions. Briefly, $5 \times 10^7$ cells were
suspended with 400 μl of the supplied solutions (MaxCyte Electroporation Buffer)
and transfected with 50 μg of linearized vector. Following transfection, cells were
incubated at 37 °C and 5% $CO_2$ for 20 min to recover. Cells were then resuspended
to $\sim 5 \times 10^5$ cells per ml in culture medium and incubated for 24 h.

**Generation of human *POLQ*-knockout cell lines.** The human *POLQ* targeting
vector pPOLQ 2A-Hyg was transfected into wild-type and $LIG4^{-/-}$ Nalm-6
cells[37] and hygromycin-resistant colonies were isolated and expanded to prepare
genomic DNA. Gene-targeting events were screened by PCR analysis using primers
POLQ 5′ ext (5′-CTGAGAGACAGCATCGACAAATGG-3′) and Universal primer
2A (5′-CACCGCATGTTAGAAGACTTCCTC-3′). Subsequently, pPOLQ 2A-Puro
was transfected into $POLQ^{+/-}$ and $LIG4^{-/-}POLQ^{+/-}$ cells, and puromycin-
resistant clones were subjected to PCR analysis using primers POLQ-N Fw
(5′-GTGCAGCTATTGAGCTCAGCAACC-3′) and POLQ-N Rv (5′-GATGCCA
AACGTAAGACGCTTCTG-3′). The disruption of the *POLQ* gene was further
confirmed by PCR analysis using primers POLQ 5′ext and POLQ exon 17 Rv
(5′-ACCACCAAGGTGTCATCACAACC-3′), and by RT–PCR analysis using
primers POLQ 5′ext and POLQ-N Rv for *POLQ*, and GAPDH RT Fw (5′-GCTGG
CGCTGAGTACGTCGTGGAG-3′) and GAPDH RT Rv (5′-CTTACTCCTTGG
AGGCCATGTGGG-3′) for *GAPDH*. RNA extraction was performed using TRIzol
reagent (Life Technologies) according to the manufacturer's instructions. To
restore *LIG4* expression in $LIG4^{-/-}POLQ^{-/-}$ cells, an EF1α promoter-driven
*LIG4* expression vector[61] (kindly provided by Dr Yoshihisa Matsumoto) was
linearized with ScaI and transfected into $LIG4^{-/-}POLQ^{-/-}$ cells, which were
then selected with 0.4 μg ml$^{-1}$ puromycin. After a 2-week cultivation, the resulting
colonies were picked and subjected to western blot analysis.

**Western blot analysis.** Cells were rinsed twice with PBS$^-$ and scraped in 200 μl
of lysis buffer (50 mM Tris-HCl (pH 6.8), 2% sodium dodecylsulfate, 10% glycerol,
100 μM dithiothreitol, 1 mM phenylmethylsulfonyl fluoride) containing Protease
Inhibitor Cocktail (1/10 volume of buffer; Sigma-Aldrich). The lysates were allowed
to stand for 20 min at 4 °C and, after sonication, centrifuged for 30 min at 16,000 g.
The supernatants were collected and used for western blot analysis. Twenty
micrograms of the lysates were electrophoresed in a 7.5% polyacrylamide gel and
then transferred onto a polyvinylidene difluoride membrane (Millipore). The
membrane was probed with anti-Lig4 antibody (1:500, kindly provided by
Dr Hirobumi Teraoka)[37] or anti-Actin antibody (1:2,000, A2066, Sigma-Aldrich).
Signals were detected with Clarity Western ECL Substrate (Bio-Rad) and analysed
using a Fuji Image Analyzer LAS-1000UVmini (Fuji Film Co.). Full scans of the
blots are shown in Supplementary Fig. 2.

**Luciferase assay.** To construct a NanoLuc Luciferase (Nluc) expression vector,
the open reading frame of a gene encoding *Nluc* was excised from pNL1.1[Nluc]
vector (Promega) and a 0.9 kb BamHI/NheI fragment containing *Nluc* gene was
inserted into BamHI/NheI-digested pIRES (Clontech), thus yielding pCMV-Nluc.
Transfection of the pCMV-Nluc vector was performed using GTE-1 electropora-
tion, Nucleofection, or MaxCyte transfection. After a 24 h incubation, an aliquot
($5 \times 10^4$ cells) was suspended in 50 μl of growth medium and luciferase activity was
measured using the Nano-Glo Luciferase Assay System (Promega) according to the
manufacturer's instructions.

**Flow cytometric analysis.** Cells were transfected with pmaxGFP (Lonza) by either
GTE-1 electroporation, Nucleofection or MaxCyte transfection. After a 24 h
incubation, green fluorescent protein (GFP)-positive cells were enumerated using a
JSAN cell sorter (Bay Bioscience).

**RI and gene-targeting assays.** For RI assays, pPGKneo[62] was linearized and
transfected into Nalm-6 cells. After cultivation for 2–3 weeks, the resulting

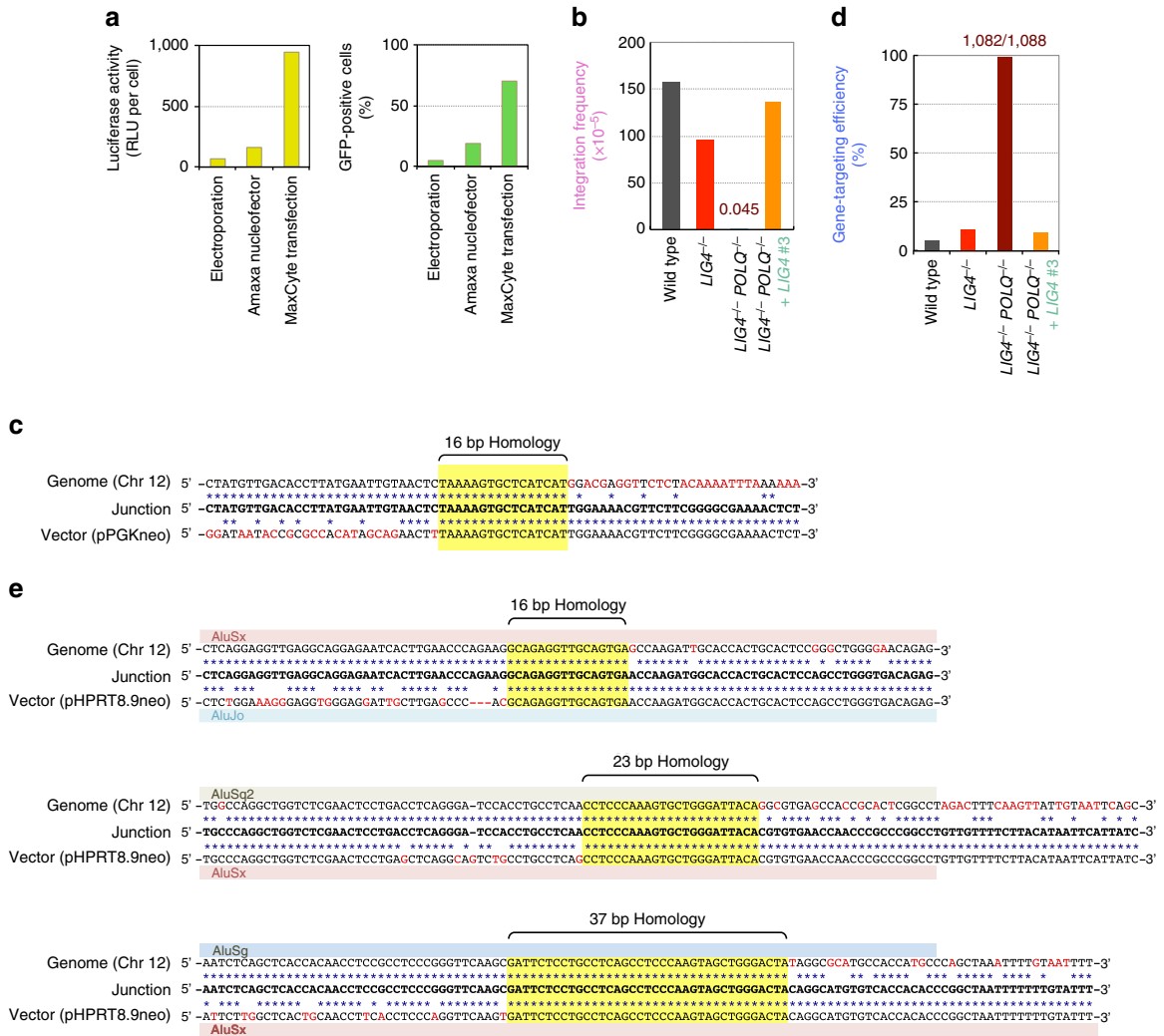

**Figure 4 | Increased transfection efficiency uncovers homology-mediated RI events. (a)** Comparison of transfection efficiency between the three transfection methods. Transient expression assays were performed using a luciferase (left) or GFP (right) expression vector (see Supplementary Fig. 2l,m). **(b)** Integration frequency of MaxCyte transfected pPGKneo in wild-type, $LIG4^{-/-}$, $LIG4^{-/-}POLQ^{-/-}$ and LIG4-complemented $LIG4^{-/-}POLQ^{-/-}$ cells. **(c)** Junction sequence in a rare RI clone obtained from $LIG4^{-/-}POLQ^{-/-}$ cells. **(d)** Gene-targeting efficiency, targeted integration frequency, and RI frequency of MaxCyte transfected pHPRT8.9-Neo2( − ) in wild-type, $LIG4^{-/-}$, $LIG4^{-/-}POLQ^{-/-}$ and LIG4-complemented $LIG4^{-/-}POLQ^{-/-}$ (#3) cells. **(e)** Junction sequences in rare RI clones obtained from $LIG4^{-/-}POLQ^{-/-}$ cells.

G418-resistant colonies were counted and the total integration frequency was calculated by dividing the number of G418-resistant colonies with that of surviving cells. For gene-targeting assays, each targeting vector was linearized and transfected into Nalm-6 cells. After a 2–3 week incubation, genomic DNA was isolated from drug-resistant colonies and subjected to PCR analysis. The gene-targeting efficiency was calculated by dividing the number of targeted clones with that of drug-resistant clones analysed. The targeted integration frequency was calculated by multiplying the total integration frequency by the targeting efficiency. The RI frequency was calculated by subtracting the targeted integration frequency from the total integration frequency.

**Junction analysis of RI clones.** The p4.5 HPRT-2A-EGFP-2A-Puro targeting vector was transfected into Nalm-6 wild-type, $LIG4^{-/-}$ or $POLQ^{-/-}$ cells and the resulting RI clones (50 clones from wild type, 25 clones from $LIG4^{-/-}$ and 16 clones from $POLQ^{-/-}$ cells) were subjected to junction analysis based on inverse PCR. Briefly, 10 μg of genomic DNA extracted from RI clones was digested with 30 U of HindIII (for 5′-junction analysis) or BamHI and BclI (for 3′-junction analysis). The digested DNA was precipitated with ethanol and 1 μg of the DNA was self-circularized in 400 μl of ligation buffer at 16 °C overnight by using T4 DNA Ligase (Takara Bio). The self-ligated DNA was extracted with phenol:chloroform and then with chloroform alone. Subsequently, the self-ligated DNA was precipitated with ethanol and used as a template for PCR amplification. PCR reactions were performed with Tks Gflex DNA Polymerase (Takara Bio) or EmeraldAmp PCR Master Mix (Takara Bio). Primers used to amplify 5′-junctions

were as follows: iPCR-EGFP Rv, 5′-TGAACTTGTGGCCGTTTACGTCGC-3′; iPCR-5-HPRT1 Fw, 5′-TCACATCTCGAGCAAGACGTTCAG-3′; iPCR-5-HPRT2 Fw, 5′-CAGAGTCCCACTATACCACAATG-3′; iPCR-5-HPRT3 Fw, 5′-CACCT-GACGTCTAAGAAACC-3′. Primers used to amplify 3′-junctions were as follows: iPCR-EGFP Fw, 5′-AACGAGAAGCGCGATCACATGGTC-3′; iPCR-3-HPRT1 Rv, 5′-CAGACTGAAGAGCTATTGTG-3′; iPCR-3-HPRT2 Rv, 5′-GCATCA-CAACATTGACACTGTGGATG-3′; iPCR-3-HPRT3 Rv, 5′-GGGCTATGCAAG-GAAGATATACTG-3′. The PCR products were cloned into the pTAKN2 T-Vector (BioDynamics Laboratory Inc.) to determine the sequence (Eurofins Genomics K.K.). BLAST programmes were used to map the junction sequences onto the human genome (University of California Santa Cruz's Genome Browser). Sequence alignments were performed using a computer software GENETYX-MAC (Software Development Co.). Sequence features (microhomologies and template insertions involving direct or inverted repeats) were identified by visual inspection. It is noteworthy that we only regarded ≥6 bp direct or inverted repeats as traits of templated insertion and hence <6 bp repeats were not taken into account as those can be frequently seen around the junctions and we could not fully rule out the possibility that <6 bp repeats at the junctions were not associated with templated insertion. Thus, the numbers of templated insertion could be even underestimated in our analysis. Furthermore, those clones that were picked independently but had identical sequences at each junction were regarded as the same recombinant cell line. Likewise, those clones in which only one side of the junctions was determined (including those showing vector-vector joining (concatemerization)) were not taken into account in the analysis, because it was unclear whether or not the inserted nucleotides relied on templated insertions using the other side of the junctions.

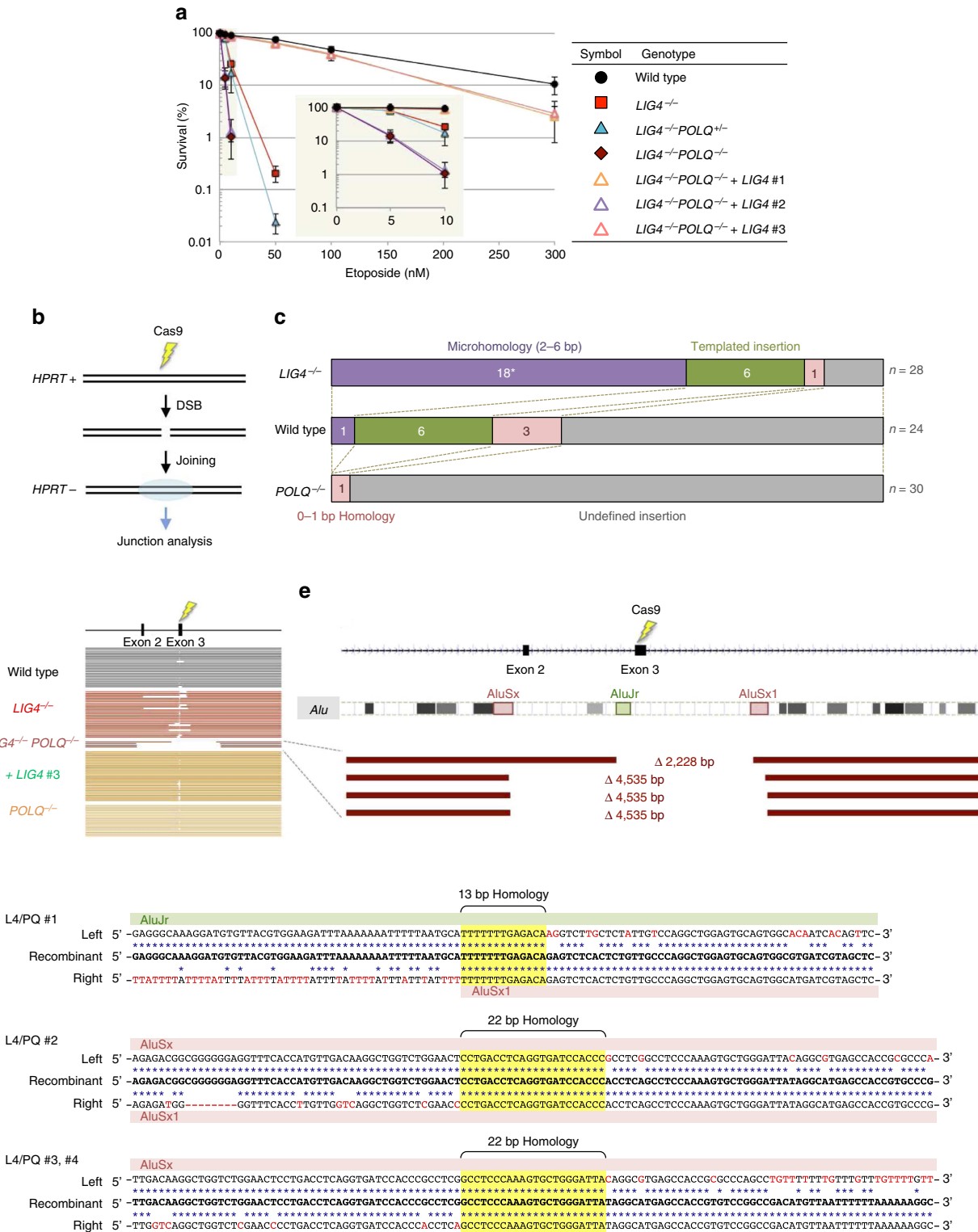

**Figure 5 | Impact of loss of Pol θ and Lig4 on chromosomal DSB repair. (a)** Cellular sensitivity to etoposide. A clonogenic survival assay was performed with different concentrations of etoposide, a topoisomerase II inhibitor that induces chromosomal DSBs (mean ± s.d.; $n = 3$). Inset graph: Enlargement at low concentrations of etoposide. **(b)** Scheme for a Cas9-induced DSB joining assay. After DSB induction at the *HPRT* locus (exon 3), 6-thioguanine-resistant colonies (that is, *HPRT*-negative cells) were subjected to junction analysis. **(c)** Features and distribution of junctions obtained from wild-type, *LIG4*−/− and *LIG4*-complemented *LIG4*−/−*POLQ*−/− (#3) cells. The features were classified into four categories; microhomology (2–6 bp homology), templated insertion (≥6 bp direct or inverted repeats), undefined insertion (1–12 bp) and 0–1 bp homology. A Fisher's exact test was used to determine statistical significance. (*$P < 0.00001$). Although not indicated, the absence of templated insertion in *POLQ*−/− cells is statistically significant ($P = 0.0052$). **(d)** Schematic representation of recombinants obtained from wild-type, *LIG4*−/−, *LIG4*−/−*POLQ*−/−, *LIG4*-complemented *LIG4*−/− *POLQ*−/− (#3) and *POLQ*−/− cells. **(e)** Schematic representation of four recombinants obtained from *LIG4*−/−*POLQ*−/− cells. The locations of *Alu* elements at the *HPRT* locus are also indicated. **(f)** Junction sequences in the four recombinants obtained from *LIG4*−/−*POLQ*−/− cells.

**Drug sensitivity assay.** To examine cellular sensitivity to etoposide, clonogenic survival assays were performed by plating exponentially growing cells, at a density of $10^2$–$10^6$ cells per 60 mm dish, into agarose medium containing various concentrations of etoposide (BioVision, Inc.). After a 2–3 week incubation at 37 °C, visible colonies were counted and the per cent survival was determined by comparing the number of surviving colonies to untreated controls.

**Chromosomal DSB joining assay.** An all-in-one CRISPR vector, pX330-U6-Chimeric_BB-CBh-hSpCas9 (pX330, Addgene plasmid #42230), was used to express Cas9 and single-guide RNA. Sense and antisense oligonucleotides of each single-guide RNA targeting sequence were annealed and cloned into pX330 at the BbsI site. The CRISPR-Cas9 target sequences (20 bp target and 3 bp PAM sequence (underlined)) used in this study include: pHPRT CRISPR1, 5′-GATGTGATGA AGGAGATGGGAGG-3′; pHPRT CRISPR2, 5′-AGTCCTACAGAAATAAAAT CAGG-3′. After transfection, cells were cultured for 1 week and then selected with 20 μM 6-thioguanine. The resulting drug-resistant (HPRT-negative) colonies were isolated and expanded to prepare genomic DNA. To exclude spontaneous mutants irrelevant to Cas9 cleavage, PCR analysis was performed using two sets of primers HPRT-F (5′-TGAGGGCAAAGGATGTGTTACGTG-3′) and HPRT US Rv (5′-CAGTCCTACAGCCCTAAAATCAGG-3′), and HPRT DS Fw (5′-GAGATGT GATGAAGGAGATGGGAG-3′) and HPRT 0.7 Rv (5′-TCCACAGTGTCAATG TTGTGATGC-3′). Subsequently, flanking sequences were PCR amplified with primers HPRT-F and HPRT 0.7 Rv or HPRT 2.0 Fw (5′-CAGCAGCTGTTCTG AGTACTTGCT-3′) and HPRT 2.3 Rv (5′-ACAGTATATCTTCCTTGCATAG CC-3′) or HPRT 3.8 Fw (5′-CACATCACAGGTACCATATCAGTG-3′) and HPRT 2.8 Rv (5′-CAGGGTAGAAATGCTACTTCAGGC-3′). PCR reactions were performed with ExTaq DNA polymerase (Takara Bio) or EmeraldAmp PCR Master Mix (Takara Bio) and PCR products were subcloned into the pGEM-T Easy vector (Promega) to determine the sequence (Eurofins Genomics K.K.). Sequence alignments were performed using a computer software GENETYX-MAC (Software Development Co.). Sequence features (microhomologies and template insertions involving direct or inverted repeats) were identified by visual inspection and defined as above (cf. junction analysis of RI clones). During data analysis, the pCRISPR HPRT2 plasmid was found to be significantly less efficient in cleaving DNA than pCRISPR HPRT1. For this reason, those recombinants that retained the pCRISPR HPRT1 target site were excluded from the data sets. Furthermore, those clones that were picked independently but had identical junction sequences (for example, recombinants #3 and #4 from $LIG4^{-/-}POLQ^{-/-}$ cells) were regarded as independent clones and thus all the recombinants analysed are included in the data sets shown in Fig. 5 and Supplementary Data 4.

**Data availability.** The data supporting the findings of this study are available within the article and its Supplementary Information, or from the authors upon reasonable request.

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

## Acknowledgements

We thank H. Teraoka for providing us with anti-Lig4 antibody and Y. Matsumoto for a *LIG4* expression vector. We also thank M. Aoki, K. Okamura, H. Watabe, Y. Abe, S. Morimoto, M. Yamagishi, M. Yoshikawa, K. Ogawa, M. Sawamura and A. Yamashita for technical assistance, and A. Kurosawa for discussions. This work was supported by grants from the Ministry of Education, Culture, Sports, Science and Technology (MEXT) (15H04323), and Yokohama City University (N.A.).

## Author contributions

S.S. performed and analysed the experiments. R.M. participated in chromosomal DSB repair assays and analysed data. N.A. designed the study, analysed data and wrote the manuscript with help from all authors.

## Additional information

**Competing interests:** The authors declare no competing financial interests.

