## [Peer review file · Nature Communications]

Reviewers' comments:

Reviewer #1 (Remarks to the Author):

This is a nice study by Saito et al., that uncovers the role of Lig4 and Polq during random integration. The process of random integration (RI), which hampers efficient gene targeting, is poorly study. Here the authors link RI events to Lig4 and polQ. The manuscript is concise, with an important observation that will impact DNA repair field and have great implication on gene targeting. I have a number of concerns that needs to be addressed prior to publication.

A major concern is related to the inconsistency between the results related to the effect of Ligase 4 on the frequency of RI (figure 2) and that presented by the accompanying manuscript. Here, the authors note a decrease in RI in the absence of Lig4, which the manuscript by Zelensky et al., show no effect upon inhibition of C-NHEJ. The message of both are papers is simple: decreased RI in the absence of C-NHEJ and PolQ, and therefore it is important to address the inconsistency in the major conclusion drawn from these studies.

Also, on this manuscript, the authors do not analyze single PolQ KO cells. Instead, they analyze PolQ; Lig4 DKO rescued with Lig4.

Second, it would be important to provide additional insight into the mechanism of action of PolQ. Is the role of PolQ during RI related to its function during replication, or linked to its impact on MMEJ. Finally, are RI events linked to a specific window of the cell cycle?

Reviewer #2 (Remarks to the Author):

For many years, efficient gene targeting in mammalian cells has been difficult due to a high proportion of illegitimate events where the targeting vector seems to randomly integrate into the genome. Previous studies have shown that loss of the classical end-joining protein LIG4 has little effect on random integration (RI). Now, Saito and colleagues demonstrate that many of these RI events proceed via polymerase theta-mediated alt-EJ. Importantly, they achieve nearly one-hundred percent efficient gene targeting in the absence of both LIG4 and POL θ (with a few SSA-mediated RI observed in experiments with high transfection efficiencies), a result which will be of great utility to researchers doing gene targeting in mice. Furthermore, they show that POL θ -mediated alt-EJ is less important in the context of a CRISPR-Cas9 induced DSB, suggesting that the use of POL θ is different in the context of two-ended DSBs vs. RI of a targeting vector.

Overall, the authors have gone to great lengths to detect random integration in the absence of both c-NHEJ and alt-EJ. Furthermore, the results are novel and resolve a long-standing issue in the field of mammalian gene targeting. I have several major issues plus a number of minor questions and clarifications that should be addressed to maximize the impact of this study.

Major issues:

1. The authors look at the junctions of RI events in the POLQ knockout cells, but they do not give values for the gene targeting efficiency or the targeted integration frequency in the POLQ $-/-$ cells. While they are expected to be similar to values observed in the LIG4 $-/-$, POLQ $-/-$, +LIG4 cells, this should be confirmed.

2. The number of RI junctions in the POLQ $-/-$ knockout cells is very small (n=6). The authors need to examine more junctions in order to make the claim that microhomologies and templated insertions are not observed in the absence of POL θ (these could come either from POLQ $-/-$ knockout cells or the LIG4 $-/-$, POLQ $-/-$, +LIG4 complemented cells).

3. The statistical probability of obtaining the junction type distribution needs to be calculated, both for the RI experiments and the CRISPR experiments shown in Figure 4C.

4. At the end of the article, the authors propose that "simultaneous inhibition of Pol θ and NHEJ will provide a novel strategy for ultra-efficient precise gene editing combined with or without nucleases." Given the moderate effect of POL θ on DSB repair efficiencies in the CRISPR experiments, the authors should include an experiment where they test the efficiency of gene targeting at the HPRT locus in the presence of a CRISPR-induced break in POLQ $-/-$ and POLQ $-/-$, LIG4 $-/-$ cells.

Minor points:

5. Page 4 line 91: specify the cell type used to create POLQ-knockout cells

6. Page 5 second paragraph: similar results (greatly reduced efficiency of DSB repair and recovery of large deletions) have previously been observed in *C. elegans* and *Drosophila* (Chan et al., 2010; Koole et al., 2014).

7. Figure 1e: for the L4 #9R junction, it isn't clear to me what the second solid-line rectangle is representing.

8. Most of the figure sizes need to be increased in size (I had to view them at 200% just to read the sequences).

9. Page 12 lines 365-368: I don't understand this sentence: "Also, those clones that were picked independently but had identical junction sequences should be regarded as the same cell line (e.g., recombinants #3 and #4 from LIG4 $-/-$ -POLQ $-/-$ cells), but were not treated as such, because the same thing should also be true for spontaneous HPRT-negative clones." Please clarify.

Reviewer #3 (Remarks to the Author):

An excellent ms. describing the importance of pol theta and the classical NHEJ pathways in controlling RI. The experiments are well-designed and test the questions nicely. These results will be highly cited.

Minor issues:

In. 173: It might be nice to cite early work in thalassemia research that shows that Alu-Alu recombination events can lead to global gene deletions.

In. 68: To test this **hypothesis**, we...

In. 104: Consistent with this **expectation**, RI...

In. 128: This **result** indicates...

Responses to Reviewers' Comments

Reviewer #1:

This is a nice study by Saito et al., that uncovers the role of Lig4 and Polq during random integration. The process of random integration (RI), which hampers efficient gene targeting, is poorly study. Here the authors link RI events to Lig4 and polQ. The manuscript is concise, with an important observation that will impact DNA repair field and have great implication on gene targeting. I have a number of concerns that needs to be addressed prior to publication.

Thank you very much for the favorable comments.

A major concern is related to the inconsistency between the results related to the effect of Ligase 4 on the frequency of RI (figure 2) and that presented by the accompanying manuscript. Here, the authors note a decrease in RI in the absence of Lig4, which the manuscript by Zelensky et al., show no effect upon inhibition of C-NHEJ. The message of both are papers is simple: decreased RI in the absence of C-NHEJ and PolQ, and therefore it is important to address the inconsistency in the major conclusion drawn from these studies.

Because Lig4 loss abolishes RI in PolQ KO cells, it is clear that C-NHEJ does have a contribution to RI. Moreover, earlier studies in mouse and avian cells have revealed a significant decrease in RI when Lig4 is ablated (J. Biol. Chem. 276:9742, 2001; Nucleic Acids Res. 36:6333, 2008). We speculate that the difference between these studies and the accompanying manuscript comes from some fundamental issues such as intrinsic NHEJ capacity or cell cycle distribution of the cells. More specifically, mouse ES cells have a low G1 population (the doubling time is ~12 hours) and weak DNA-PK activity, perhaps rendering the contribution of NHEJ smaller. Alternatively or additionally, if the size of the vector used is small, the contribution from NHEJ becomes smaller, as we have shown in this study as well as in previous work.

Also, on this manuscript, the authors do not analyze single PolQ KO cells. Instead,

they analyze PolQ; Lig4 DKO rescued with Lig4.

In the revised manuscript, we have presented the data of single PolQ KO cells, which do display essentially the same phenotypes as the Lig4-complemented DKO cells.

Second, it would be important to provide additional insight into the mechanism of action of PolQ. Is the role of PolQ during RI related to its function during replication, or linked to its impact on MMEJ. finally, are RI events linked to a specific window of the cell cycle?

We are actually interested in those important issues, which must be solved in future studies. We believe that PolQ's role in RI is directly related to alternative end-joining (which comprises MMEJ and insertion-based joining events, as described in the manuscript), though other possibilities could be pursued, particularly in light of "when RI occurs in the cell cycle". **[Redacted]** Because now we know that the only two mechanisms, PolQ-dependent end-joining and C-NHEJ, are responsible for RI events, it will be important to determine when during the cell cycle each RI can take place.

Reviewer #2:

For many years, efficient gene targeting in mammalian cells has been difficult due to a high proportion of illegitimate events where the targeting vector seems to randomly integrate into the genome. Previous studies have shown that loss of the classical end-joining protein LIG4 has little effect on random integration (RI). Now, Saito and colleagues demonstrate that many of these RI events proceed via polymerase theta-mediated alt-EJ. Importantly, they achieve nearly one-hundred percent efficient gene targeting in the absence of both LIG4 and POL θ (with a few SSA-mediated RI observed in experiments with high transfection efficiencies), a result which will be of great utility to researchers doing gene targeting in mice.

Furthermore, they show that POL θ -mediated alt-EJ is less important in the context of a CRISPR-Cas9 induced DSB, suggesting that the use of POL θ is different in the context of two-ended DSBs vs. RI of a targeting vector.

Overall, the authors have gone to great lengths to detect random integration in the absence of both c-NHEJ and alt-EJ. Furthermore, the results are novel and resolve a long-standing issue in the field of mammalian gene targeting. I have several major issues plus a number of minor questions and clarifications that should be addressed to maximize the impact of this study.

Thank you very much for the favorable comments.

Major issues:

1. The authors look at the junctions of RI events in the POLQ knockout cells, but they do not give values for the gene targeting efficiency or the targeted integration frequency in the POLQ $-/-$ cells. While they are expected to be similar to values observed in the LIG4 $-/-$, POLQ $-/-$, +LIG4 cells, this should be confirmed.

In the revised manuscript, we have presented the data of single PolQ KO cells, which exhibit very similar gene targeting efficiency and targeted integration frequency to that of Lig4-complemented DKO cells.

2. The number of RI junctions in the POLQ $-/-$ knockout cells is very small (n=6). The authors need to examine more junctions in order to make the claim that microhomologies and templated insertions are not observed in the absence of POL θ (these could come either from POLQ $-/-$ knockout cells or the LIG4 $-/-$, POLQ $-/-$, +LIG4 complemented cells).

We have presented more RI junctions (n=24), which led us to confirm that templated insertions and >2-bp microhomologies are never observed in the absence of POL θ , and junctions with 0-1 bp homology become prominent instead.

3. The statistical probability of obtaining the junction type distribution needs to be calculated, both for the RI experiments and the CRISPR experiments shown in Figure 4C.

In the revised manuscript, we have presented statistical data on the

junction type distribution, and now it is clear, in particular, that 2-6 bp microhomology and 0-1 homology are highly specific to A-EJ and NHEJ, respectively, in both the RI experiments and the CRISPR experiments (shown in Figures 3a and 5c, respectively).

4. At the end of the article, the authors propose that “simultaneous inhibition of Pol θ and NHEJ will provide a novel strategy for ultra-efficient precise gene editing combined with or without nucleases.” Given the moderate effect of POL θ on DSB repair efficiencies in the CRISPR experiments, the authors should include an experiment where they test the efficiency of gene targeting at the HPRT locus in the presence of a CRISPR-induced break in POLQ $-/-$ and POLQ $-/-$, LIG4 $-/-$ cells.

We have removed “combined with” and changed the sentence to “simultaneous inhibition of Pol θ and NHEJ will provide a novel strategy for ultra-efficient gene targeting even without using artificial nucleases.”

We respectfully think that the CRISPR-induced gene targeting, albeit interesting and important, is beyond the scope of this article and thus we prefer not to present our data in the present manuscript. **[Redacted]**

Minor points:

5. Page 4 line 91: specify the cell type used to create POLQ-knockout cells.

Now specified.

6. Page 5 second paragraph: similar results (greatly reduced efficiency of DSB repair and recovery of large deletions) have previously been observed in *C. elegans* and *Drosophila* (Chan et al., 2010; Koole et al., 2014).

We have cited the important *Drosophila* paper. (The *C. elegans* work

does not seem to include data of a PolQ/NHEJ double mutant.) Thank you for the suggestion.

7. Figure 1e: for the L4 #9R junction, it isn't clear to me what the second solid-line rectangle is representing.

We have added the junction sequence of the other (opposite) side of the integration site to make this figure clearer. In addition, in order to indicate the fact that such type of TI-mediated RI is not rare, we have presented an additional figure (Figure 3c), which shows that TI events not only rely on a genome or vector sequence present nearby but also can utilize a sequence around the opposite junction of the integration site.

8. Most of the figure sizes need to be increased in size (I had to view them at 200% just to read the sequences).

Now enlarged where needed.

9. Page 12 lines 365-368: I don't understand this sentence: "Also, those clones that were picked independently but had identical junction sequences should be regarded as the same cell line (e.g., recombinants #3 and #4 from LIG4-/-POLQ-/- cells), but were not treated as such, because the same thing should also be true for spontaneous HPRT-negative clones." Please clarify.

We have changed the sentence to provide a clearer explanation. The point was that clones with identical junction sequences were regarded as independent clones and treated as such in the data sets.

Reviewer #3:

Minor issues:

In. 173: It might be nice to cite early work in thalassemia research that shows that Alu-Alu recombination events can lead to global gene deletions.

Thank you for the comment. We have cited the relevant paper on page 9 (ref. 53).

In. 68: To test this **hypothesis**, we...

In. 104: Consistent with this **expectation**, RI...

In. 128: This **result** indicates...

All corrected.

REVIEWERS' COMMENTS:

Reviewer #1 (Remarks to the Author):

The authors have addressed my concerns. I am in favor of publication.

Reviewer #2 (Remarks to the Author):

The authors have addressed all of my major concerns. The inclusion of the POLQ $-/-$ experiments, the increased number of junctions in cells lacking POLQ, and the inclusion of statistical analyses further bolster their claims. The expanded introduction and the reorganization of figures/increased figure size also improves the manuscript. Figure 5 is still small and should be increased in size upon publication.

My only concern now is that the authors may be overstating some of their results in the revised text. Examples of this include:

1. "It may be worth mentioning that the absolute frequency of targeted integration is somewhat elevated in POLQ-knockout cells, an observation compatible with recent reports that suggest a role for Pol θ in suppressing HR."

To me, this statement is not supported by the data.

2. "In this study, we have shown that A-EJ is defined, at least genetically, as a single mechanism that exclusively requires Pol θ . Thus, the NHEJ and A-EJ pathways explain all non-homologous DSB repair in human cells."

While the authors have shown that random integration events with longer MH and template insertions are Pol θ -dependent, they have not shown that A-EJ is a single mechanism. The literature is replete with many examples of DSB repair junctions involving non-templated insertions that do not require Pol θ . I recommend that these two sentences be removed.

Response to Reviewers' Comments

Reviewer #1:

The authors have addressed my concerns. I am in favor of publication.

We appreciate this reviewer's help to improve the quality of our manuscript.

Reviewer #2:

The authors have addressed all of my major concerns. The inclusion of the POLQ -/- experiments, the increased number of junctions in cells lacking POLQ, and the inclusion of statistical analyses further bolster their claims. The expanded introduction and the reorganization of figures/increased figure size also improves the manuscript. Figure 5 is still small and should be increased in size upon publication.

We appreciate this reviewer's help to improve the quality of our manuscript.

We believe that Figure 5 is now large enough to be legible upon publication.

My only concern now is that the authors may be overstating some of their results in the revised text. Examples of this include:

1. "It may be worth mentioning that the absolute frequency of targeted integration is somewhat elevated in POLQ-knockout cells, an observation compatible with recent reports that suggest a role for Pol θ in suppressing HR."

To me, this statement is not supported by the data.

We have removed the sentence.

2. "In this study, we have shown that A-EJ is defined, at least genetically, as a single mechanism that exclusively requires Pol θ . Thus, the NHEJ and A-EJ pathways explain all non-homologous DSB repair in human cells."

While the authors have shown that random integration events with longer MH and template insertions are Pol θ -dependent, they have not shown that A-EJ is a single

mechanism. The literature is replete with many examples of DSB repair junctions involving non-templated insertions that do not require Pol θ . I recommend that these two sentences be removed.

We have removed the two sentences.